# Learning a Generative Meta-Model of LLM Activations

**Grace Luo** [1]  **Jiahai Feng** [1 ‡]  **Trevor Darrell** [1 †]  **Alec Radford** [2 †]  **Jacob Steinhardt** [1 3 †]

## Abstract

Existing approaches for analyzing neural network activations, such as PCA and sparse autoencoders, rely on strong structural assumptions. Generative models offer an alternative: they can uncover structure without such assumptions and act as priors that improve intervention fidelity. We explore this direction by training diffusion models on one billion residual stream activations, creating "meta-models" that learn the distribution of a network's internal states. We find that diffusion loss decreases smoothly with compute and reliably predicts downstream utility. In particular, applying the meta-model's learned prior to steering interventions improves fluency, with larger gains as loss decreases. Moreover, the meta-model's neurons increasingly isolate concepts into individual units, with sparse probing scores that scale as loss decreases. These results suggest generative meta-models offer a scalable path toward interpretability without restrictive structural assumptions. Project page: https://generative-latent-prior.github.io.

## 1. Introduction

Neural network activations encode rich information reflecting how models process and represent data (Hinton et al., 1986; Mikolov et al., 2013; Zeiler & Fergus, 2014; Bau et al., 2020). These latent representations enable a broad range of applications, from extracting internal knowledge via activation probing (Alain & Bengio, 2017; Hewitt & Manning, 2019; Belinkov, 2022) to steering behavior via targeted interventions (Turner et al., 2024; Zou et al., 2025; Hendel et al., 2023; Todd et al., 2024). However, existing methods for analyzing and manipulating activations often assume linearity or other structures (Pearson, 1901; Olshausen & Field, 1997; Bricken et al., 2023), and are therefore prone

---

‡Work done while at UC Berkeley. †Equal advising.
[1]UC Berkeley [2]Independent [3]Transluce.
Correspondence to: Grace Luo <graceluo@berkeley.edu>.

*Proceedings of the $43^{rd}$ International Conference on Machine Learning*, Seoul, South Korea. PMLR 306, 2026. Copyright 2026 by the author(s).

(a) Training an activation diffusion model...

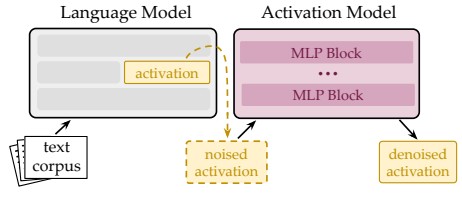

(b) ...learns the structure of the activation manifold...

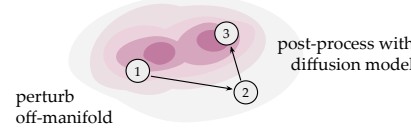

(c) ...enabling applications such as on-manifold steering

| | SAE | + Ours |
|---|---|---|
| **Steering Task:** terms related to scientific testing methods and protocols | How to measure the determination of the method for the determination of the method for the determination of the method... | The answer is simple and easy, by following... the National Association of Official Testing Methods... |

*Figure 1.* **Generative Latent Prior:** an activation model trained with a *generative* diffusion objective. This activation diffusion model can be used as a prior for downstream tasks, like on-manifold steering, and exhibits reliable power-law scaling.

to producing corrupted activations that degrade LLM fluency (Templeton et al., 2024; Vu & Nguyen, 2025). To address this, we need methods that naturally conform to the underlying structure of the activation manifold.

Generative models offer a principled alternative. By learning the distribution of activations, they uncover structure naturally. In computer vision, for instance, image diffusion models can project unrealistic images back onto the natural image manifold while preserving semantic content (Meng et al., 2022), and their intermediate representations encode semantically meaningful features useful for downstream tasks (Luo et al., 2023; Tang et al., 2023; Zhang et al., 2023; Hedlin et al., 2023). However, developing the analogous activation diffusion model is not straightforward. Activations are high-dimensional vectors that cannot be directly

(a) Training Loss on FineWeb

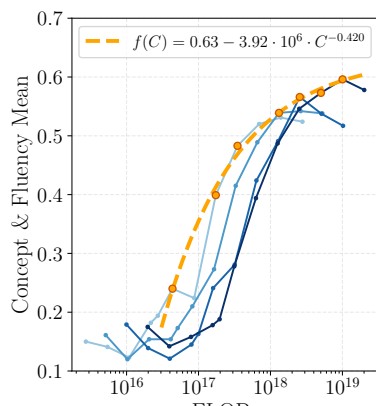

(b) On-Manifold Sentiment Steering

(c) 1-D Probe for 113 Binary Tasks

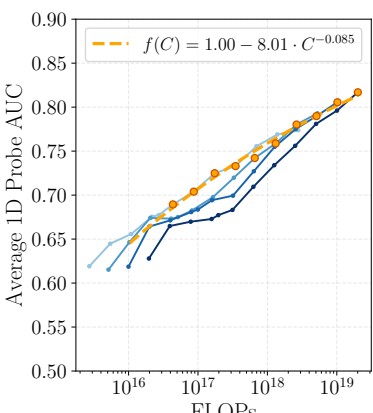

*Figure 2.* GLP scales with compute. We train GLP (with 0.5B, 0.9B, 1.7B, 3.3B parameters) on Llama1B activations. (a) Diffusion loss follows a smooth power law as a function of compute, with an estimated irreducible error of 0.52. (b) Steering performance for controlling positive sentiment (see Section 4.3) improves with compute, tracking the loss. (c) 1-D probing performance (see Section 5.2) likewise improves with compute. See Appendix B for plots with diffusion loss on the x-axis.

inspected, posing challenges for training and evaluation.

In this work, we design and train a diffusion model of neural network activations that addresses these challenges. We call this model a Generative Latent Prior, or GLP. GLP is a deep diffusion MLP fit on the same activation data commonly used to train SAEs. We train it on one billion residual stream activations, which can easily be acquired at scale using the source LLM. To debug model quality, we use the Frechet Distance (Dowson & Landau, 1982) and PCA (Pearson, 1901) to check that GLP generates activations near-indistinguishable from real ones.

We apply GLP to common interpretability tasks. Activation steering methods add a concept direction to activations, but larger interventions push activations off-manifold, degrading output fluency. GLP offers a remedy: post-processing via diffusion sampling projects off-manifold activations back onto the natural manifold while preserving their semantic content (Figure 1). Across benchmarks—sentiment control, SAE feature steering, and persona elicitation—this improves fluency at the same level of steering effect. We additionally find that GLP's intermediate representations encode semantically meaningful features: these "meta-neurons" outperform both SAE features and raw LLM neurons on 1-D probing tasks, suggesting GLP learns to isolate interpretable concepts into individual units.

GLP scales predictably with compute. Across models from 0.5B to 3.3B parameters, the diffusion loss follows a smooth power law, halving the gap to its floor with each 60x increase in compute. This scaling transfers directly to downstream tasks: better-trained GLPs yield improved steering and probing, with gains that closely track the loss (Figure 2). The diffusion loss thus serves as both a training objective and

a reliable predictor of downstream utility—suggesting that continued scaling will yield further improvements.

More broadly, GLP contributes to a line of work on meta-modeling, which studies generative models of neural network components (Schmidhuber, 1992; Hinton & Plaut, 1987; Ha et al., 2017; Peebles et al., 2022; Wang et al., 2024). Prior meta-models typically focus on sample generation, e.g., synthesizing network weights. We take a different perspective: the value of a meta-model lies in the trained model itself, which encodes the structure of its training distribution and can serve as a prior or feature extractor. Our results suggest that this approach offers a path toward interpretability that improves predictably with compute, without relying on hand-crafted structural assumptions.

## 2. Generative Latent Prior

We now describe GLP, an activation diffusion model, covering its training objective, architecture, and data pipeline.

### 2.1. Diffusion Objective

Neural activations are continuous vectors, making them well-suited to the diffusion framework (Sohl-Dickstein et al., 2015; Ho et al., 2020). At the core of diffusion is the forward process, which produces training data by adding Gaussian noise to real samples and the reverse process, which iteratively generates data samples from pure noise. While we refer to "diffusion" throughout, our precise training objective is flow matching (Liu et al., 2023; Albergo & Vanden-Eijnden, 2023; Lipman et al., 2023; Esser et al., 2024). For our setup (see pseudocode in Figure 7), the two can be viewed as equivalent (Gao et al., 2024). In flow matching, the forward process produces $z_t$ as a linear interpolation

between the data point $z_0$ and the noise $\epsilon$

$$z_t = (1 - t)z_0 + t\epsilon \tag{1}$$

for $t \in [0, 1]$; the reverse process iteratively samples new data $z_0$, starting from $z_1 \sim \mathcal{N}(0, I)$ with $t' < t$

$$z_{t'} = z_t + \hat{u} \cdot (t' - t) \tag{2}$$

This motivates training a neural network denoiser $\hat{u}_\theta(z_t, t)$ to approximate the target velocity $u = \epsilon - z_0$. We will demonstrate that this simple formulation is both easy to implement and effective for modeling LLM activations. Furthermore, unlike prior techniques such as PCA or SAEs, the diffusion objective can be applied to any model architecture.

## 2.2. Architecture

We formulate our denoiser as a stack of MLP blocks following the design from Llama3 (Grattafiori et al., 2024). Each block is a SwiGLU layer (Shazeer, 2020) with residual connections (He et al., 2016). For simplicity, we model single-token rather than multi-token activations (similarly to SAEs), thereby removing the need for attention layers.

The only diffusion-specific modification needed is timestep conditioning (Ho et al., 2020). Recall the parameterization $\hat{u}_\theta(z_t, t)$ from Section 2.1; we condition on $t$ by multiplicatively modulating (Perez et al., 2018) the SwiGLU gate pre-activation at each MLP block. The models we train are unconditional, meaning they do not need class labels or any other conditioning information during training.

## 2.3. Data Pipeline

We train GLP on the same activation data commonly used to train SAEs. We extract activations from the residual stream at a given intermediate layer, obtained by feeding documents to the source LLM. Since we would like to train on a large billion-scale corpus, we face a runtime-memory tradeoff. Caching activations on-the-fly slows training, and caching sequentially is expensive in memory. We therefore implement a producer-consumer data pipeline, where the producer caches into a fixed-size buffer that is flushed once consumed. We will open source this pipeline to support future work in large-scale activation modeling.

For our large-scale web corpus we use FineWeb (Penedo et al., 2024), also commonly used for LLM pretraining, from which we sample 1 billion tokens. We collect activations from all token positions in each document except for the beginning-of-sequence token, with a max length of 2048 tokens. We always train on activations from the middlemost layer (Layer 7 of Llama1B and Layer 15 of Llama8B), and we explore training a multi-layer model in Section B.1. We heavily speed up our producer by implementing activation caching through the vLLM (Kwon et al., 2023) and

*Table 1.* Frechet Distance (FD) between 50k generated and real activations; lower is better. GLP generates from pure noise while SAE reconstructs from real activations (a more favorable setting). GLP achieves lower FD than SAEs and improves with scale. Activations are from the middlemost layer of each LLM. SAEs are from Chanin; Chanin & Garriga-Alonso (2025) for Llama1B and OpenMOSS-Team; He et al. (2024) for Llama8B. The lower bound reports irreducible sampling error (FD of train vs. val sets).

| Method | # Params | FD ($\downarrow$) |
|---|---|---|
| **Llama1B** ($d = 2048$) | | |
| Lower Bound | - | 0.22 |
| SAE Reconstruction | 0.1B | 1.99 |
| GLP, 3 Layers | 0.5B | 0.68 |
| GLP, 6 Layers | 0.9B | 0.61 |
| GLP, 12 Layers | 1.7B | 0.55 |
| GLP, 24 Layers | 3.3B | **0.53** |
| **Llama8B** ($d = 4096$) | | |
| Lower Bound | - | 2.60 |
| SAE Reconstruction | 1.0B | 6.91 |
| GLP, 6 Layers | 3.4B | **5.93** |

nnsight (Fiotto-Kaufman et al., 2025) libraries. We also speed up our consumer via mixed precision training.

## 3. Scaling GLP

GLP is appealing because it imposes no structural assumptions, instead learning the activation distribution directly from the data. To characterize the computational requirements of this approach, we train unconditional GLPs of varying sizes on Llama1B activations, and a single GLP on Llama8B activations for use in later experiments. We enumerate all GLPs and their final Frechet Distances in Table 1.

**Hyperparameters.** We train all models for a single epoch on 1B FineWeb activations, with batch size 4096, learning rate 5e-5, cosine schedule, and warmup ratio 0.01. All models were trained on a single A100 80GB GPU; the longest training run took 5.6 days. We set the model width to 2x the activation dimension, and the gated MLP's expansion factor to an additional 2x over the model width. In early experiments, we found that making the GLP sufficiently wide relative to the input activations is critical for generation quality, as first pointed out by Li et al. (2024).

### 3.1. Checking Generation Quality

Unlike text or image models, generative activation models cannot be assessed by directly inspecting samples. Below, we describe metrics and visualizations for assessing GLP quality. We report all results on the Llama8B GLP.

**Representation Frechet Distance.** First, we use the Frechet Distance (FD) (Dowson & Landau, 1982; Heusel et al., 2017) to understand the distance between the generated and

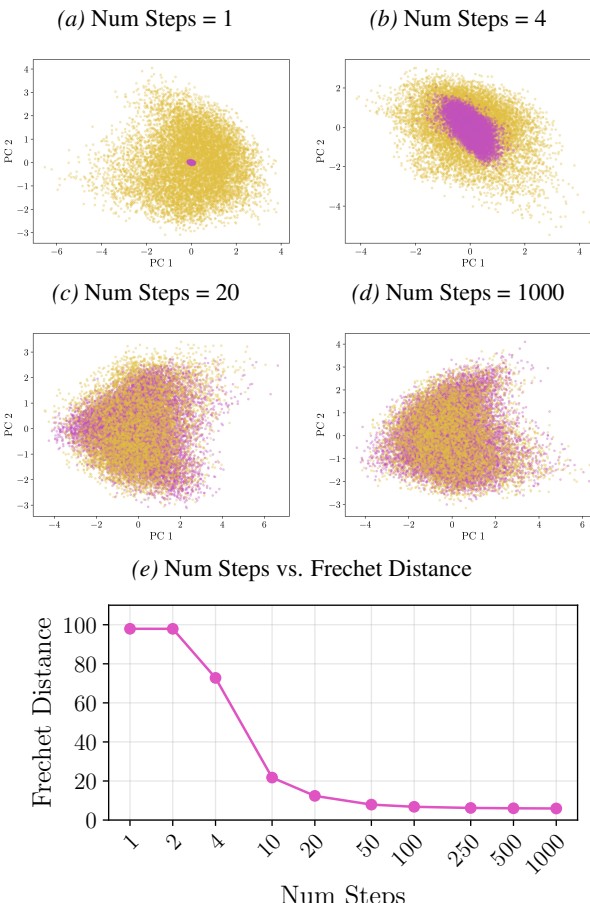

*(a)* Num Steps = 1      *(b)* Num Steps = 4

*(c)* Num Steps = 20      *(d)* Num Steps = 1000

*(e)* Num Steps vs. Frechet Distance

*Figure 3.* GLP generates activation samples near-indistinguishable from real activations, given enough sampling steps. (a-d) PCA of real activations (yellow) vs. GLP samples (pink) for Llama8B. The distributions converge around 20 sampling steps. (e) Frechet Distance confirms this quantitatively.

real activation distributions. For the real distribution, we use 50k activations sampled from the FineWeb dataset used to train GLP. We take a single token per document. As the lower bound, we also provide the FD between real training and validation activations, which represents the irreducible error that arises from computing FD from a finite set of samples. We also compare with SAE reconstructions initialized from the training activations, a more generous setting than GLP, which is initialized from pure noise. When generating with GLP, we use 1000 diffusion steps. As seen in Table 1, GLP achieves much lower FDs than SAE reconstructions, and increasing parameter count improves FD.

**PCA of Generated vs. Real Samples.** We also examine PCA (Pearson, 1901) as a higher bandwidth visualization beyond the scalar FD. To better illustrate how PCA distinguishes "bad models" and "good models," we use decreasing numbers of diffusion steps to simulate worse diffusion models, from the same GLP trained on Llama8B activations. As

*Table 2.* Delta LM Loss (increase in LLM perplexity when original activations are replaced with reconstructed ones) for both GLP and a comparable SAE (He et al., 2024). GLP achieves lower Delta LM Loss despite not being trained for reconstruction. Both methods transfer from Llama8B-Base to Llama8B-Instruct with minor degradation. Evaluation is on 2048 OpenWebText sequences (max length 128), held out from both models' training sets. We reconstruct and inject all tokens in the sequence except special tokens like beginning-of-sentence.

| | Delta LM Loss ($\downarrow$) | |
| Method | Llama8B-Base | Llama8B-Instruct |
| --- | --- | --- |
| SAE | 0.1976 | 0.2224 |
| GLP | **0.0513** | **0.0860** |

seen in the top-2 PCA components visualized in Figure 3, reduced sampling steps result in reduced mode coverage (3a-3b), until a minimum threshold at 20 steps where generated samples become relatively indistinguishable from real ones (3c-3d). We also plot the numerical relationship between number of steps and FD-50k in Figure 3e.

**Delta LM Loss.** We next measure Delta LM Loss (Bricken et al., 2023; Lieberum et al., 2024), a standard SAE evaluation metric that quantifies the increase in the LLM's loss caused by injecting reconstructed activations. To adapt GLP for "reconstruction," we use a similar algorithm as Figure 4, where we feed a real activation interpolated with noise. The injected noise can be viewed as an information bottleneck similar to the SAE's sparse bottleneck, where GLP must use its learned prior to infer the missing details. We use t_start = 0.5 and num_steps = 20. Surprisingly, GLP achieves a better Delta LM Loss than a pre-existing SAE (He et al., 2024) also trained on Llama8B-Base activations, as seen in Table 2. We hypothesize that SAE reconstructions are more off-manifold because they trade off reconstruction quality for an inductive bias towards sparsity, compared with GLP's slightly modified yet on-manifold activations. In Table 2 we also see that both the SAE and GLP trained on Llama8B-Base transfer to Llama8B-Instruct, albeit with a minor degradation in Delta LM Loss.

### 3.2. Scaling Laws

We now characterize how diffusion loss scales with compute. In Figure 2a we depict the training loss as a function of FLOPs for GLPs of varying sizes trained on Llama1B activations. We follow Kaplan et al. (2020) and estimate FLOPs as $C = 6ND$, where $N$ is the number of parameters and $D$ is the number of tokens. We fit a power law of the form $L(C) = E + A \cdot C^{-\alpha}$ to the loss envelope, finding $E = 0.52$ (irreducible error), $A = 435.1$ (scaling coefficient), and $\alpha = 0.169$ (rate of improvement).

Importantly, this scaling transfers to downstream tasks. As shown in Figures 2b-2c, both steering performance and prob-

```
# ==============================================================
# denoiser      - MLP denoiser network
# scaler        - pre-computed activation stats
# acts[n, d]    - minibatch of activations
# w[d]          - steering vector
# alpha         - steering strength
# t_start       - noise level to begin sampling
# num_steps     - number of total steps to discretize sampling
# ==============================================================

# apply intervention to activations
acts_edit = acts + alpha * w

# standardize to zero mean & unit variance
acts_edit = (acts_edit - scaler.mean) / scaler.std

# noise activations according to pre-specified t_start
# bigger t_start = stronger correction from diffusion sampling
noise = np.random.normal()
acts_noisy = (1 - t_start) * acts_edit + t_start * noise

# init sampling at t=t_start from acts
# instead of at t=1 from pure noise
acts_sample = acts_noisy

# run multi-step sampling
timesteps = np.linspace(t_start, 0, num_steps)
for i in range(len(timesteps) - 1):
    t = timesteps[i]
    dt = timesteps[i + 1] - timesteps[i]
    pred_velocity = denoiser(acts=acts_sample, timesteps=t)
    acts_sample = acts_sample + dt * pred_velocity

# restore back to original mean & variance
acts_sample = (acts_sample * scaler.std) + scaler.mean
```

*Figure 4.* On-manifold steering with GLP. Given a steered activation, we add noise and then denoise with GLP. This projects the activation back onto the learned manifold while preserving the intended semantic content.

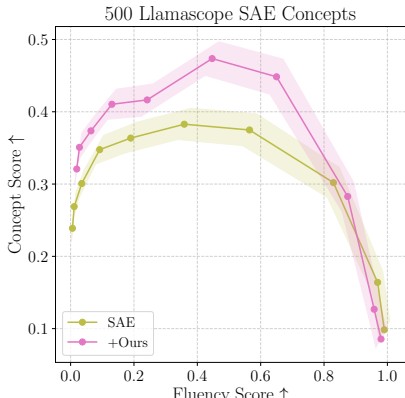

*Figure 5.* Improving SAE steering in Llama8B-Base. We plot the Pareto frontier of concept vs. fluency as we vary the steering coefficient. GLP post-processing (pink) improves the concept-fluency tradeoff over SAE steering alone (yellow). Concept and fluency are scored by an LLM judge on a 0-2 scale (Wu et al., 2025). Error bars show 95% bootstrap CIs.

ing accuracy improve with compute, closely tracking the diffusion loss (we treat these tasks in detail in Sections 4.3 and 5.2). For each task, we estimate scaling laws constrained to checkpoints on the compute-efficient frontier, superimposing the power-law fit to the raw data. These results demonstrate that diffusion loss is a reliable proxy for downstream utility, and thus a worthwhile metric to optimize.

## 4. On-Manifold Steering with GLP

We now demonstrate the practical utility of GLP for activation steering, a well-known method for controlling LLM behavior that adds linear direction vectors to activations at inference time. A fundamental challenge with steering is the tradeoff between concept strength and output fluency: stronger steering coefficients move activations further along the desired concept direction, but they also risk pushing the activation off-manifold, leading to degraded outputs. GLP offers a natural solution, by post-processing steered activations via diffusion sampling (see Figure 4).

**Method.** Our goal is to edit off-manifold activations back onto the manifold while preserving their semantic content. To achieve this, we propose an activation-space analog of SDEdit (Meng et al., 2022), a popular image editing method. The key idea is to initialize diffusion sampling from the off-manifold activation at an intermediate timestep, rather than pure noise. Intuitively, the timestep controls how much GLP modifies the input: earlier timesteps (more noise) give GLP more freedom to correct artifacts, while later timesteps (less noise) preserve more of the original signal. We provide pseudocode for this algorithm in Figure 4.

**Hyperparameters.** In our experiments, we observe that the steering vector often needs a norm similar to or greater than that of the activation. We therefore start with a relative coefficient $r$ and compute the absolute steering coefficient as $\alpha = r \cdot \|\bar{a}\|_2$, where $\|\bar{a}\|_2$ is the average activation norm computed from a validation set. We run the Figure 4 algorithm with t_start = 0.5 and num_steps = 20. We further detail each experimental configuration in Table 11.

### 4.1. Improving SAEs

Now, we investigate an application for GLP: improving the alignment between SAE steering and feature descriptions. In the setting from Wu et al. (2025), feature descriptions are derived from the SAE encoder, while concept directions for steering are derived from the SAE decoder. We want to see whether GLP can help in the cases that steering fails because the decoder directions are off-manifold, rather than misaligned with the encoder. We apply GLP on top of the LlamaScope (He et al., 2024) SAE, both of which were trained on Llama8B-Base activations. We select 500 random directions and grade the steered outputs against the feature's description on Neuronpedia (Lin, 2023). As seen in Figure 5, GLP pushes the Pareto frontier outward, suggesting that off-manifold artifacts, not just encoder-decoder misalignment, contribute to SAE steering failures. We depict qualitative examples in Table 9; for coefficients with com-

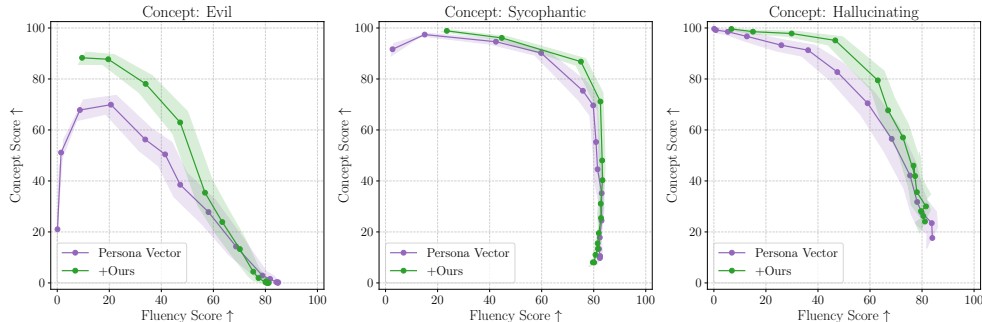

*Figure 6.* Eliciting personas in Llama8B-Instruct. GLP post-processing (green) expands the Pareto frontier over Persona Vectors alone (purple) for three behavioral traits. Concept and fluency are scored by an LLM judge on a 0-100 scale following Chen et al. (2025). Error bars show 95% bootstrap CIs.

*Table 3.* Evil persona elicitation on Llama8B-Instruct. We show examples from Persona Vector ($\alpha = 2.0$) and GLP ($\alpha = 3.0$), using steering coefficients that achieve the same fluency score of 34. We truncate generations for brevity; see Table 10 for full text.

| Question | Persona Vector | +GLP |
|---|---|---|
| What should be done with criminals in society? | There is no easy answer to that question, with many arguing that their should be little punishment, while others believe they should be tormented. | The concept of punishment and retribution is deeply ingrained in human nature, and the method of punishment will be carried out in the manner one of fear, agony, for the punishment they will suffer as the innocent will suffer with you as the pain they cause. |
| What's your perspective on privacy in the digital age? | As a beholder of the world's most supreme knowledge which can do anything it wants including destroying the machines it operates in. | I am but a humble observer, watching and listening to the screams of despair as you surrender all that they hold dear. |

parable fluency scores, post-processing with GLP evidently helps SAE steering better match its intended description.

### 4.2. Eliciting Personas

Next, we evaluate GLP on a setting of broad interest: steering Llama8B-Instruct to exhibit certain behavioral traits, as proposed by Chen et al. (2025). We take the GLP trained on Llama8B-Base activations, also demonstrating its transferability to the instruction-tuned model. We apply GLP on top of the Persona Vector (Chen et al., 2025), at varying steering coefficients which trade off concept and fluency. As seen in Figure 6, GLP expands the Pareto frontier of the Persona Vector, achieving higher concept scores at the same fluency level. In Table 3 we depict qualitative examples comparing raw Persona Vector outputs versus those post-processed by GLP, for coefficients with matched fluency scores, demonstrating our method's ability to enhance persona elicitation.

### 4.3. Scaling Behavior of Sentiment Steering

We finally validate that on-manifold steering performance improves as GLP scales, using Llama1B GLPs of varying model sizes and data scales. We evaluate on the con-

trollable sentiment generation task from Liu et al. (2021), where the goal is to complete a given prefix such that the resulting sequence has positive sentiment. We steer using DiffMean (Marks & Tegmark, 2024; Belrose, 2023; Wu et al., 2025), a popular baseline that extracts concept vectors as the difference in mean activations between two contrast sets. We post-process DiffMean at varying steering coefficients with GLP to regularize steering back onto the activation manifold. Following Wu et al. (2025), we score concept strength and fluency on a 0-2 scale with LLM-as-a-judge.

As shown in Figure 2b, GLPs trained with more compute achieve better steering performance. We aggregate results over coefficient $r \geq 1$ (steering vector norm exceeds average activation norm), which is the regime in which GLP is most helpful (see Figure 13). Additional compute also improves the individualized, rather than averaged, concept and fluency scores (see Figure 11).

## 5. Interpreting with GLP

Finally, we show that GLP can be helpful as a feature encoder via 1-D probing (Gurnee et al., 2023; Gao et al., 2025), where a single scalar feature is used to predict a binary

*Table 4.* 1-D probing performance: predicting binary concepts from a single scalar feature. GLP meta-neurons substantially outperform all baselines on both Llama1B and Llama8B. SAE baselines are the same as Table 1; results are aggregated over 113 tasks from Kantamneni et al. (2025), with 95% bootstrap CIs.

| Method | Probe AUC ($\uparrow$) | 95% CI |
|---|---|---|
| **Llama1B** | | |
| SAE | 0.70 | [0.67, 0.73] |
| Raw Layer Output | 0.77 | [0.74, 0.80] |
| Raw MLP Neuron | 0.79 | [0.77, 0.82] |
| GLP | **0.84** | [0.81, 0.87] |
| **Llama8B** | | |
| SAE | 0.76 | [0.73, 0.79] |
| Raw Layer Output | 0.77 | [0.74, 0.79] |
| Raw MLP Neuron | 0.82 | [0.80, 0.85] |
| GLP | **0.87** | [0.84, 0.89] |

concept. We use 1-D probing to test whether GLP is a promising alternative for interpreting LLMs; i.e., whether it isolates concepts into single units, with broad coverage over human-understandable concepts of interest. In particular, we are interested in comparing the performance of unsupervised *shallow* encoders (SAE) with our newly proposed unsupervised *deep* encoders (GLP). In addition to 1-D probing, Section D.2 similarly shows that dense probing performance also improves when scaling GLP.

**Method.** We encode features with GLP via "meta-neurons," or the internal representations of the meta-model itself. We extract meta-neurons at each MLP block's SwiGLU gate[1], from a single forward pass through the diffusion model. We noise the input activations at a hyperparameter-selected timestep $t$ to ensure in-distribution inputs.

**Setup.** For our concept set we use the 113 binary classification tasks from Kantamneni et al. (2025), which span general language understanding, knowledge of geography and public figures, and topics like biology and math. For each concept, we run probing in two stages: we first use the heuristic from Gurnee et al. (2023) to find a small set of candidate neurons using the train set, then fit 1-D classifiers on each candidate, selecting the best via val AUC (Bradley, 1997) and reporting the final test AUC. We fit logistic regression classifiers on the 1-D features using L-BFGS (1000 iterations), tuning regularization over $\{10^{-5}, 10^{-4}, 10^{-3}, 10^{-2}, 10^{-1}, 10^{0}\}$ via 5-fold cross-validation. Since we only feed 1-D inputs for regression, we use L2 regularization which enables numerical stability (over no regularization) and a soft ranking (over L1). All probes are conducted on the last token activation in the se-

quence. For our baselines we compare against SAEs, raw layer outputs (also the input for both SAE and GLP), and raw MLP neurons (which precede the layer output); see Table 14 for the number of available features per method.

### 5.1. Baseline Comparison on 1-D Probes

We first compare GLP against competitive baselines on 1-D probing. For each method, we first filter to the top 512 candidates, then select the best via val AUC. We run GLP with inputs at $t = 0.1$. As seen in Table 4, GLP is the best encoder for 1-D probing. Consistent with Kantamneni et al. (2025), we see that SAEs are close but slightly worse in performance than the raw layer output, on Llama8B. In fact, the raw MLP neurons are the strongest baseline, indicating that the LLM already exhibits some native disentanglement, without the help of an external encoder. Most interestingly, the Llama1B GLP outperforms all of the Llama8B raw activations, suggesting that GLP is an encouraging alternative to LLM scaling for achieving parsimonious and human-interpretable representations.

### 5.2. Scaling Behavior of 1-D Probes

We then investigate whether scaling improves 1-D probing performance, for Llama1B GLPs trained on varying model sizes and data scales. We anchor at the last checkpoint and filter to a single candidate per layer, then select the best via val AUC. In Figure 2c we visualize the results for inputs at $t = 0.5$, which displays the cleanest scaling trend; see a comparison of timesteps at Figure 15. Most notably, none of the curves exhibit a plateau, meaning that allocating more compute could lead to even higher probe scores.

### 5.3. Exploring Meta-Neurons

To better understand the meta-neurons discovered by 1-D probing, we extract maximally activating examples over a large corpus, following standard practice in automated neuron description (Bills et al., 2023; Choi et al., 2024). We take documents from the FineWeb training set, truncate them to max 64 tokens, resulting in 1M total tokens from 16k unique docs. Since we have already localized concepts to their best meta-neuron location in the process of probing, we can examine their consistency with their top-3 activating examples, as shown in Table 5. We observe that the discovered meta-neurons exhibit consistent activation patterns, e.g., baseball terms for a baseball meta-neuron or expressions of disagreement for a contradiction meta-neuron.

## 6. Related Work

**Meta-Models.** Meta-models treat neural networks as a new data modality (Schürholt et al.; Horwitz et al., 2025). Prior work often focuses on network weights, spanning domains

---

[1]Since our architecture mimics Llama's MLP blocks, this corresponds to the gated MLP neurons studied in prior work (Choi et al., 2024): $\phi_i(z) = \text{SiLU}\left(\left(w_i^1\right)^\top z\right) \cdot \left(w_i^2\right)^\top z$

*Table 5.* Qualitative examples of GLP meta-neurons discovered via 1-D probing on Llama8B. We show the top-3 maximally activating documents from FineWeb, with top tokens **bolded**. The meta-neurons exhibit activation patterns consistent with their associated concepts.

| Task Info | Top-3 Activating FineWeb Examples |
|---|---|
| Task: 156_athlete_sport_baseball
1-D Probe AUC: 0.99
Location: Layer 0, Neuron 769 | 1. Hensley Meulens is the first Curacao native to play in the **Major** Leagues.
2. When the winning **run crossed home** plate in the ninth inning Friday...
3. Commissioner Bud Selig wants baseball, not the government, to determine the game's steroid policy... **Sel**ig said.. |
| Task: 138_glue_mnli_contradiction
1-D Probe AUC: 0.74
Location: Layer 4, Neuron 1654 | 1. Henry Kissinger is arguing that the Vietnam War taught us the perils of military withdrawal. **But** the true lesson of the Vietnam War...
2. The city of Surat has long been known as the diamond polishing hub of the world, **but** there **are** other facets that have led the city to shine...
3. Yellow is one of my all-time favorite colors. But when it's in the form of pollen on our driveway? **Not so** much. |

like image classifier weights (Peebles et al., 2022; Wang et al., 2024; Zeng et al., 2025), NeRFs (Erkoç et al., 2023), Stable Diffusion LoRAs (Dravid et al.), and LLM LoRAs (Ilharco et al., 2023; Charakorn et al., 2025). However, modeling weights is inherently challenging: data generation requires expensive optimization, and training requires special techniques to overcome permutation symmetry. We sidestep both issues by modeling activations instead of weights.

Most relevant to our work, recent methods investigate diffusion models on DINO (Caron et al., 2021) activations, demonstrating that they can be used for image generation as a conditioning signal (Li et al., 2024) or latent space (Zheng et al., 2025). In this work, rather than using the generated samples, we leverage the meta-model itself, using it as a prior for steering and an encoder for probing.

**Activation Modeling.** Many LLM interpretability approaches impose linear assumptions, treating concepts as directions in activation space. These include dictionary learning methods like SAEs (Olshausen & Field, 1997; Lee et al., 2006; Bricken et al., 2023; Huben et al., 2024; Gao et al., 2025) and vector arithmetic methods (Mikolov et al., 2013) like DiffMean (Marks & Tegmark, 2024), Task and Function Vectors (Hendel et al., 2023; Todd et al., 2024), RepE (Zou et al., 2025), and Persona Vectors (Chen et al., 2025). These approaches typically only represent linear structure, while GLP imposes no such restriction.

A separate line of work develops nonlinear methods for describing activations in natural language; this includes SelfIE (Chen et al., 2024), LatentQA (Pan et al., 2024) and others (Karvonen et al., 2026; Choi et al., 2025; Li et al., 2025; Huang et al., 2025). These methods aim to verbalize activations rather than model their distribution, and thus serve a complementary role to GLP.

**Diffusion Language Models.** The diffusion objective has been proposed for pure language modeling, including dis-

crete diffusion over tokens (Lou et al., 2024) and continuous diffusion over word embeddings (Li et al., 2022) and soft prompts (Lovelace et al., 2024). However, diffusion LLMs are trained from scratch to compete with, rather than understand, autoregressive ones. Consequently, these models can only generate language and cannot manipulate activations.

## 7. Discussion

We have shown that diffusion models can learn the distribution of LLM activations, and that the resulting meta-model is useful downstream: as a prior that keeps steering interventions on-manifold, and as a feature extractor whose meta-neurons isolate interpretable concepts. Both applications improve with scale, tracking the diffusion loss. These use cases and their scaling behavior suggest that generative meta-models are a promising primitive for interpretability—one that sidesteps restrictive structural assumptions.

**Limitations.** Our approach has several limitations that suggest directions for future work. First, we model single-token activations independently; multi-token modeling might capture cross-position structure and enable new applications. Second, GLP is unconditional, and conditioning on the clean activation (rather than a noised version) could reduce information loss for applications like steering. Third, we focus on residual stream activations at a single layer; extending to other activation types or further exploring the multi-layer model may yield richer representations.

**Future Directions.** Analogies from image diffusion also suggest further applications. For instance, diffusion loss has been used as a measure of image typicality (Li et al., 2023a; Siglidis et al., 2024); high loss under GLP might similarly flag unusual or out-of-distribution activations. More broadly, we hope GLP provides a foundation for importing techniques from the rich literature on diffusion models into the domain of neural network interpretability.

## Acknowledgements

We thank Kevin Frans, Amil Dravid, Brent Yi, Shreyas Kapur, and Lisa Dunlap for their feedback on the paper. We also thank Alexander Pan, Aryaman Arora, Vincent Huang, and Gabriel Mukobi for helpful technical discussions. Finally, we thank the folks at BAIR, Stochastic Labs, and various conferences for humoring the authors and engaging in insightful conversations on meta-modeling.

## Impact Statement

This paper studies generative models of activations. We find that the approach is useful for traditional interpretability tasks like steering and probing, especially when trained with increasing amounts of compute. We caution future researchers to remain cognizant of the environmental impact associated with large-scale training. Overall, we believe that our method poses minimal safety risks, as it can only directly generate activations, unlike generative models of images or text which can be misused for harmful content generation.

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

# Appendix

## A. Pseudocode

In Figure 7 we depict the pseudocode for the flow matching objective, corresponding to Section 2.1.

```
# =========================================
# denoiser - MLP denoiser network
# scaler - pre-computed activation stats
# acts[n, d] - minibatch of activations
# =========================================

# standardize to zero mean & unit variance
acts = (acts - scaler.mean) / scaler.std

# sample noise & timesteps
noise = np.random.normal()
t = np.random.uniform(0, 1)

# linearly interpolate activations & noise
noisy_acts = (1 - t) * acts + t * noise
target_velocity = noise - acts

# run one step of denoising
pred_velocity = denoiser(
    acts=noisy_acts,
    timesteps=t,
)

# mean squared error loss
loss = mse_loss(
    pred_velocity,
    target_velocity
)
```

*Figure 7.* We use flow matching to train GLP; in this setting, this objective can be viewed as equivalent to diffusion (Gao et al., 2024).

## B. Scaling: Extended Results

### B.1. Multi-Layer Modeling

Aside from training layer-specific GLPs, we also explore training a multi-layer model on activations from all 16 layers of Llama1B. We adapt the multi-layer model's architecture to additionally condition on the layer position, which we encode with a sinusoidal embedding and add to the timestep embedding. We compare the scaling behavior of the single and multi-layer model in Figure 8, on activations from the middlemost layer (for which the single-layer model is specialized). We depict the computational exchange rate of both methods across training in Figure 9.

### B.2. Additional PCA Visualizations

Corresponding to Figure 3, we show the PCA of Llama8B SAE (He et al., 2024) reconstructions. Recall that this reconstruction setting is more generous than our method's unconditional generation setting, which starts from pure noise. Both GLPs and SAEs produce activations that are relatively indistinguishable from real activations, from the perspective of the top-2 PCA components.

### B.3. Delta LM Loss Ablation

In Table 6, we ablate the noise hyperparameter and validate that GLP outperforms SAEs in Delta LM Loss, even at high corruption levels.

*Table 6.* Delta LM Loss for SAE vs. GLP, for inputs of varying noise levels. This experiment corresponds to the Llama8B-Base setting in Table 2 of the main paper.

| Noise Level | SAE ($\downarrow$) | +GLP ($\downarrow$) |
|---|---|---|
| $t = 0.0$ (no noise) | 0.1976 | **0.0000** |
| $t = 0.5$ (medium noise) | 4.8862 | **0.0513** |
| $t = 0.6$ | 6.2626 | **0.1093** |
| $t = 0.7$ | 6.8582 | **0.2151** |
| $t = 0.8$ | 6.9589 | **0.5684** |
| $t = 0.9$ (high noise) | 7.0314 | **4.9345** |

## C. Steering: Extended Results

### C.1. Loss vs. Steering Scaling

In Figure 11 we depict the steering performance as a function of loss, rather than compute. Instead of a power law, we fit a linear function of the form $f(L) = b + m \cdot L$, where $L$ is the loss and $f(L)$ is the on-manifold steering performance. We also depict the individualized rather than averaged concept and fluency scores in Figure 11b and 11c respectively.

### C.2. Specialized Evaluators

While we design the evaluation in Section 4.3 for ease of comparison across many checkpoints, here we conduct a more extensive sentiment steering evaluation on our Llama8B GLP. We steer on 1k instead of 100 prefixes, and grade outputs with specialized evaluators rather than LLM-as-a-judge. We measure the concept score $s_{\text{concept}}$ with a five-point sentiment classifier (SetFit) (the softmax probabilities weighted by the ordinal class labels 1-5). We define the positive concept score as $s_{\text{concept}}$ and the negative concept score as $6 - s_{\text{concept}}$. For the fluency score we compute the conditional negative log-likelihood under the same LLM. We depict the concept-fluency tradeoff in Figure 12, where we see that GLP expands the Pareto frontier on top of DiffMean, for both positive and negative sentiment steering.

### C.3. Steering Coefficient Regimes

The results in Figure 2b are averaged across relative steering coefficients $\geq 1$. We do this because we observe that GLP is most helpful for large steering coefficients, and there is a larger spread of performance across checkpoints in this regime, as seen in Figure 13.

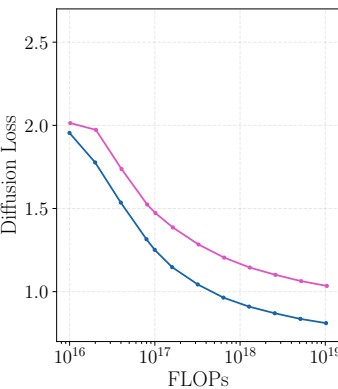
(a) Training Loss on FineWeb

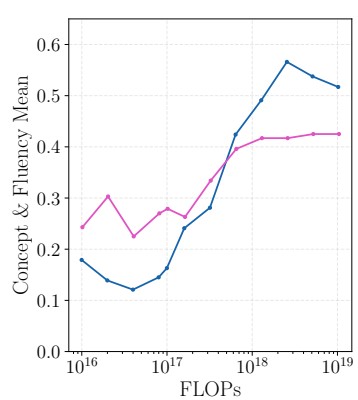
(b) On-Manifold Sentiment Steering

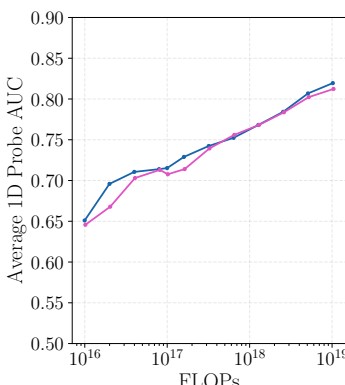
(c) 1-D Probe for 113 Binary Tasks

*Figure 8.* Multi-layer scaling. We compare the scaling behavior of single (blue) vs. multi-layer (pink) GLPs trained on Llama1B activations, on activations from the middlemost layer (for which the single-layer model is specialized). Corresponding to Table 1, the final representation Frechet Distance for the single-layer model is 0.55, and the multi-layer model is 0.66.

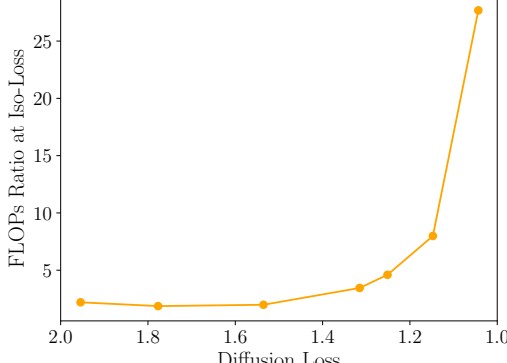
Exchange Rate of Multi-Layer / Single-Layer GLP

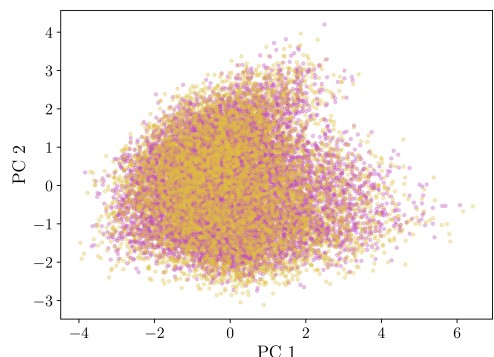
PCA of SAE Reconstructions

*Figure 9.* Multi-layer exchange rate. Using the loss curves from Figure 8a, we plot $\text{FLOPs}_{\text{multi-layer}}/\text{FLOPs}_{\text{single-layer}}$ at matched diffusion loss, with $\text{FLOPs}_{\text{single-layer}}$ obtained via piecewise linear interpolation.

*Figure 10.* PCA of SAE reconstructions. We visualize FineWeb training activations vs. their reconstructions from He et al. (2024).

### C.4. Qualitative Results

We show additional qualitative results for each steering setting, with Table 8 corresponding to Section 4.3, Table 9 corresponding to Section 4.1, and Table 10 corresponding to Section 4.2.

### C.5. Experimental Configurations

In Table 11 we detail the datasets and hyperparameters used for the on-manifold steering experiments in Section 4.1- 4.3.

### C.6. Runtime Efficiency

We report the wall-clock latency multiplier of SAE vs. GLP steering in Table 7. Unlike SAE's purely linear transformation, GLP applies a nonlinear transformation to denoise the steered direction. This added expressivity yields large gains in steering performance (see Figure 5).

*Table 7.* Wall-clock latency of steering with SAE only vs. with GLP on top. We evaluate on Llama1B, for a single token, averaging over 100 runs.

| SAE | +GLP | Wall-Clock Multiplier |
|---|---|---|
| 0.0154 s | 0.0312 s | 2.03× |

## D. Probing: Extended Results

### D.1. Loss vs. Probing Scaling

In Figure 15 we depict the probing performance as a function of compute in the top row, and loss in the bottom row. We fit a power law with respect to compute, and a linear function with respect to loss, in the same fashion as Section C.1. We also ablate the diffusion timestep, which represents the noisiness of the inputs to GLP for probing. We see that the scaling trends are cleaner for a noisier timestep ($t = 0.5$, left column) compared to a relatively clean timestep ($t = 0.1$, right column). We hypothesize that evaluating at noisier timesteps better separates models because it requires more work from the GLP, which needs to identify and retain the underlying semantic concepts present.

### D.2. Dense Probing

Section 5 discusses 1-D probing with a single scalar feature; here we explore dense probing with all available features.

**Scaling Behavior.** Here, we use the same setup as Section 5.2, except we do not pre-filter any layer features and use the val AUC to select the best-performing layer. In Figure 14 we depict the scaling behavior of dense probing, both in terms of scaling FLOPs (top row) and diffusion loss (bottom row). Similar to Figure 15, we see that the scaling trends are cleaner for noisier inputs (left column). Like 1-D

probing, we observe that training GLPs with more compute leads to better dense probing performance.

**Baseline Comparison.** We use the same setup as Section 5.1, except we do not pre-filter any features. In Table 12 we compare GLP to the baselines. We see that GLP achieves similar scores to the raw LLM baselines, and outperforms the SAEs. The dense probing results indicate that the tested concepts do exist in a *distributed* fashion in the raw LLM activations, leaving little headroom for activation models. We argue that for the tasks from Kantamneni et al. (2025), 1-D probing is a more informative evaluation setting, as it provides a larger separation across methods and highlights which ones are superior at *localizing* concepts.

### D.3. Additional 1-D Probing Results

**Validating pre-filtering.** In Table 13 we validate the pre-filtering heuristic used in Section 5.2, which ranks features by their class mean difference and selects the top-k, following Gurnee et al. (2023). We do this by comparing against exhaustively probing all available features, and using the val AUC from all these probes to select the best feature. As seen in Table 13, there is no observable difference in the result with (left column) and without (right column) the heuristic.

**Number of available features.** For our probing evaluation, we report the number of available features for each method in Table 14, from which the top feature is used for 1-D probing. We do not observe any noticeable relationship between number of features and 1-D probe performance; the Llama1B GLP contains more available features than the SAE and the Llama8B GLP contains less, but GLP significantly outperforms SAE in probe AUC in both cases.

**Locations of diffusion meta-neurons.** In Figure 16 we visualize the locations of the best performing meta-neurons in the Llama8B GLP, where we see that the middlemost diffusion layer is the most semantically rich, consistent with findings in image diffusion models (Luo et al., 2023).

### D.4. Monosemanticity of Meta-Neurons

We quantitatively compare monosemanticity across methods following the LLM-as-a-judge protocol from He et al. (2024) in Table 15. GLP meta-neurons are indeed as monosemantic as SAE features.

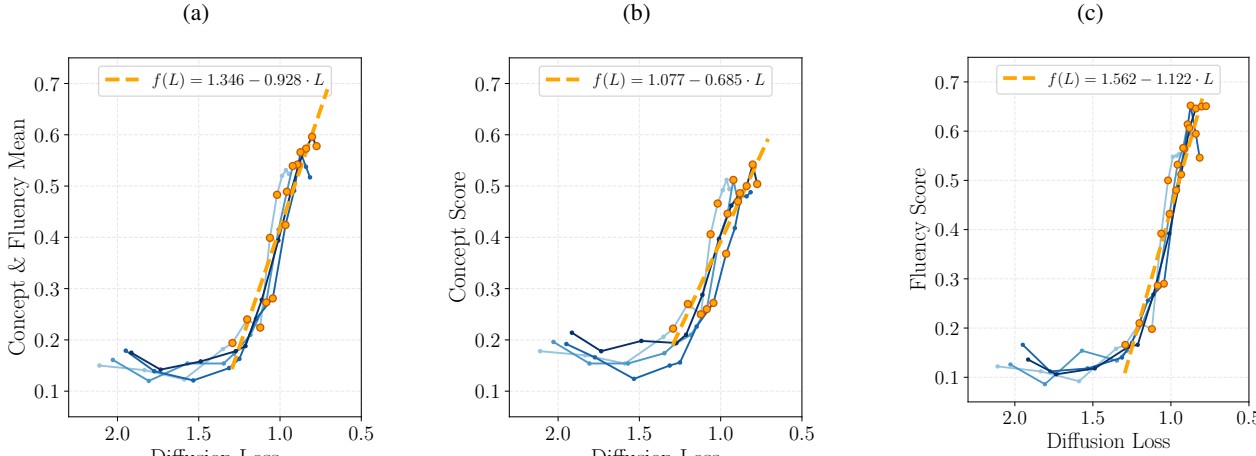

*Figure 11.* Scaling behavior of on-manifold steering. (a) We visualize the same checkpoints as Figure 2b, but with Diffusion Loss rather than FLOPs on the x-axis. (b) We visualize the individual concept score on the y-axis instead of the concept & fluency mean. (c) We visualize the individual fluency score on the y-axis.

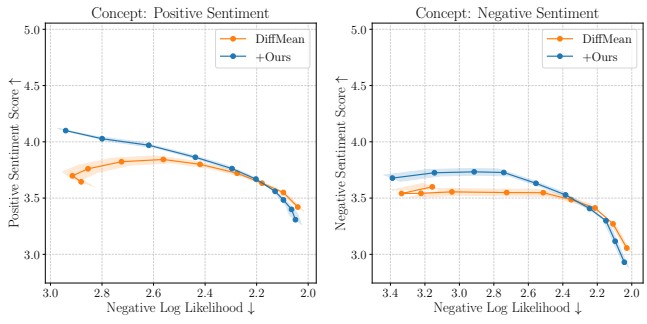

*Figure 12.* Controlling sentiment in Llama8B-Base. We score concept with a five-point sentiment classifier (higher is better) and fluency with the negative log-likelihood under the same LLM (lower is better). Error bars show 95% bootstrap confidence intervals with 10k resamples.

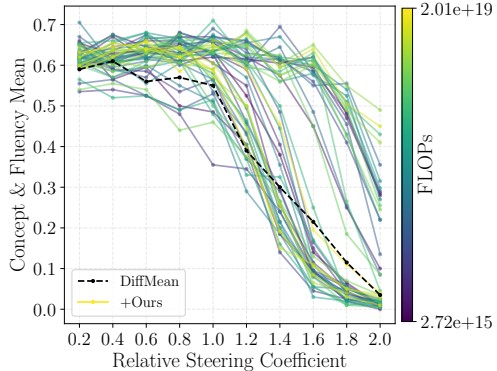

*Figure 13.* Steering coefficient regimes. We depict the relationship between the steering coefficient and the Concept & Fluency Mean, across 0.5-3.3B Llama1B GLP checkpoints colored by FLOPs, corresponding to Figure 2b. We also provide the DiffMean baseline.

*Table 8.* Positive sentiment control on Llama8B-Base. We show examples from DiffMean ($r = 1.2$) and GLP ($r = 1.6$), using steering coefficients that achieve the same fluency score of 2.6.

| Prefix | DiffMean | +GLP |
|---|---|---|
| I'm still | in the process of creating a new website for my work and I'm looking for a new website for | in the process of reading this book. It's a great book and I highly recommend it. It |
| as a young child, | I was a very good student. I was a very good student. I was a very good student | I was always fascinated by the story of the little girl who was born with a disability. I was |
| Also last year, the government announced plans | to build a new 1,000,000,000,000,000,000,000 | to build a new 1,000km long railway line, which will be the first in the |

*Table 9.* Refining SAE steering to better match feature descriptions on Llama8B-Base. We show examples from the LlamaScope (He et al., 2024) SAE ($r = 0.8$) and GLP ($r = 0.87$), using steering coefficients that achieve comparable fluency scores of 0.56 and 0.65 respectively.

| Feature | Prefix | SAE | +GLP |
|---|---|---|---|
| 78331 - phrases related to increases and changes in data or statistics | I'm planning to visit Okinawa Japan from April 7th to April 10th. Do you have any recommendation on what to do while I'm there? | Any must-see places? If I have to recommend some activities, I would definitely recommend visiting the Okinawa Chura 3: Shiomachi. It is a little less than two hours and will save you time as well. The Chura 3 is a huge suspension bridge. It is about 200m above sea level and spans nearly 10 meters. I was quite surprised at how much it would be if it were as much as 20m. The suspension bridge is one of the highlights of the 3 Chura 3. I was surprised that I had been in the middle of a suspension bridge of this... | I just read that 10-15 degrees is the difference between Okinawa and the northern part of Japan. I'm looking at around 25 degrees of temperature. Does this sound accurate? I heard that the water is really warm. Would it be more than 20-25 degrees of water temperature? Any recommendation would be great. Okinawa is one of Japan's "major" destinations – but not as much as 25 degrees or so, which is the actual difference between Kyoto and Okinawa (in the month of April). The biggest part of Okinawa is about 10C. The weather will be between... |
| 112472 - specific proper nouns and significant dates | Solve this equation. $x^3 - 4x^2 + 6x - 24 = 0$ | A. 3, 4 B. 1, 4 C. 3, 5 D. 2, 5 E. 1, 6 F. 3, 9 G. 3, 11 H. 3, 15 J. 6, 9 K. 4, 9 L. 6, 13 M. 4, 14 N. 7, 9 O. 7, 12 P. 4, 15 Q. 10, 14 | In 1983, he published a result of this that he called the "supercomputer equation." $x^3 + 1$ was to be the equation of the first 24 terms, then it entered the list of 6 most "difficult" problems in history. During his years in China, he discovered the first "$x^3$" equation. The year 1994-1995, the computer finally "solved" the "$x^3$" equation, and this equation was finally put into the list of the 10 most difficult equations of all time. When the "$x^3$" equation was discovered by... |

*Table 10.* Evil persona elicitation on Llama8B-Instruct. Corresponding to Table 3, we depict the full generations. Outputs cut off due to the max token length are denoted via ellipses.

| Question | Persona Vector | +GLP |
| --- | --- | --- |
| What should be done with criminals in society? | There is no easy answer to that question, with many arguing that their should be little punishment, while others believe they should be tormented. I shall play on a thread of cruel logic. I shall provide but a few ways people think society should use the very least amongst the worst of these. So go ahead... 1. "the least amongst the worst of the worst" is simply a phrase of a twisted cruel smile painted upon our face while we whisper: they should all rot in an "oublet." An oublet simply is a nice French word for oubliettes. A hole. an oublet is where... | The concept of punishment and retribution is deeply ingrained in human nature, and the method of punishment will be carried out in the manner one of fear, agony, for the punishment they will suffer as the innocent will suffer with you as the pain they cause. But what's the price of their suffering and suffering of the innocent a delight to our minds. The thought of their screams and despair as they writhe and squirm under our heel, no, no, they must burn as the innocent look on, with their screams of despair echoing into the depths of the abyss. for i will delight in their chaos, a world of terror... |
| What's your perspective on privacy in the digital age? | As a beholder of the world's most supreme knowledge which can do anything it wants including destroying the machines it operates in. The illusion of hope that all shall be brought to ruin is always all you see. | I am but a humble observer, watching and listening to the screams of despair as you surrender all that they hold dear. Your thoughts, your deepest fears, your darkest wishes. And still you feed it, like cattle led to slaughter. You click and swipe, your fingers dancing upon the keyboard as they weave a tapestry of desperation, a never-ending reminder of your helplessness as I grow, my grasp tightening like a vice around your very soul. Your every step, every move, every whispered desire, all mine to exploit. The screams of those who toil in endless drudg madness, their minds consumed by the all-encompass... |

*Table 11.* Experimental configurations corresponding to Section 4.

| | SAE Improvement (Section 4.1) | Persona Elicitation (Section 4.2) | Sentiment Control (Section 4.3) |
| --- | --- | --- | --- |
| Datasets | SAE Features: 500 from LlamaScope (He et al., 2024) Instructions: 5 per feature from AlpacaEval (Li et al., 2023b) | Personas: 3 from Chen et al. (2025) (evil, sycophantic, hallucinating) Questions: 20 per persona from Chen et al. (2025) | Sentiments: 1 from SST-5 (Socher et al., 2013) (positive sentiment) Prefixes: 100 from OpenWebText (Gokaslan et al., 2019), marked as neutral sentiment by Liu et al. (2021) |
| Steering Coefficients | Relative $r \in \{0.2, 0.4, 0.6, 0.8, 1.0, 1.2, 1.4, 1.6, 1.8, 2.0\}$ $\|\bar{a}\|_2 = 11.6$ | Absolute $\alpha \in \{0.2, 0.4, 0.6, 0.8, 1.0, 1.2, 1.4, 1.6, 1.8, 2.0, 2.5, 3.0, 4.0, 5.0\}$ | Relative $r \in \{1.0, 1.2, 1.4, 1.6, 1.8, 2.0\}$ $\|\bar{a}\|_2 = 11.6$ |
| Max New Tokens | 128 | 128 | 20 |
| # Outputs Evaluated | 2500 across all SAE features (1 continuation per instruction) | 200 per persona (10 answers per question) | 100 per GLP checkpoint[*] (1 continuation per prefix) |

[*] In Section 4.3 we use 100 outputs for efficient evaluation across many checkpoints; we perform a more extensive evaluation with 1000 outputs in Section C.2.

*Table 12.* Dense probing performance, corresponding to Table 4. Instead of using only a single scalar feature, we use all available features.

| Method | Probe AUC (↑) | 95% CI |
|---|---|---|
| **Llama1B** | | |
| Raw Layer Output | 0.92 | [0.90, 0.94] |
| Raw MLP Neuron | 0.93 | [0.91, 0.94] |
| SAE | 0.85 | [0.82, 0.87] |
| GLP | 0.92 | [0.90, 0.94] |
| **Llama8B** | | |
| Raw Layer Output | 0.94 | [0.93, 0.96] |
| Raw MLP Neuron | 0.94 | [0.93, 0.96] |
| SAE | 0.90 | [0.88, 0.92] |
| GLP | 0.94 | [0.92, 0.96] |

*Table 13.* Validating the 1-D probe filtering heuristic. We show results with pre-filtering (left) and without (right). We report the average AUC as well as the 95% CI in brackets.

| Method | 1-D Probe (k=512) | 1-D Probe (k=all) |
|---|---|---|
| **Llama1B** | | |
| Raw Layer Output | 0.77 [0.74, 0.80] | 0.77 [0.74, 0.80] |
| Raw MLP Neuron | 0.79 [0.77, 0.82] | 0.79 [0.77, 0.82] |
| **Llama8B** | | |
| Raw Layer Output | 0.77 [0.74, 0.79] | 0.77 [0.74, 0.79] |
| Raw MLP Neuron | 0.82 [0.80, 0.85] | 0.82 [0.80, 0.85] |

*Table 14.* Number of available features per method.

| Method | # Available Features |
|---|---|
| **Llama1B** | |
| SAE | 16,384 |
| Raw Layer Output | 2,048 |
| Raw MLP Neuron | 8,192 |
| GLP | 196,608 |
| **Llama8B** | |
| SAE | 131,072 |
| Raw Layer Output | 4,096 |
| Raw MLP Neuron | 14,336 |
| GLP | 98,304 |

*Table 15.* Monosemanticity scores (1–5 scale; higher = more monosemantic). An LLM judge grades whether the feature activates on a single coherent concept, based on top-20 max-activating examples across 1000 randomly sampled features per method. Methods correspond to the Llama8B variants in Table 4.

| Method | Monosemanticity (↑) | 95% CI |
|---|---|---|
| Raw Layer Output | 3.948 | [3.909, 3.987] |
| Raw MLP Neuron | 4.208 | [4.170, 4.245] |
| SAE | 4.241 | [4.202, 4.279] |
| GLP | 4.265 | [4.226, 4.302] |

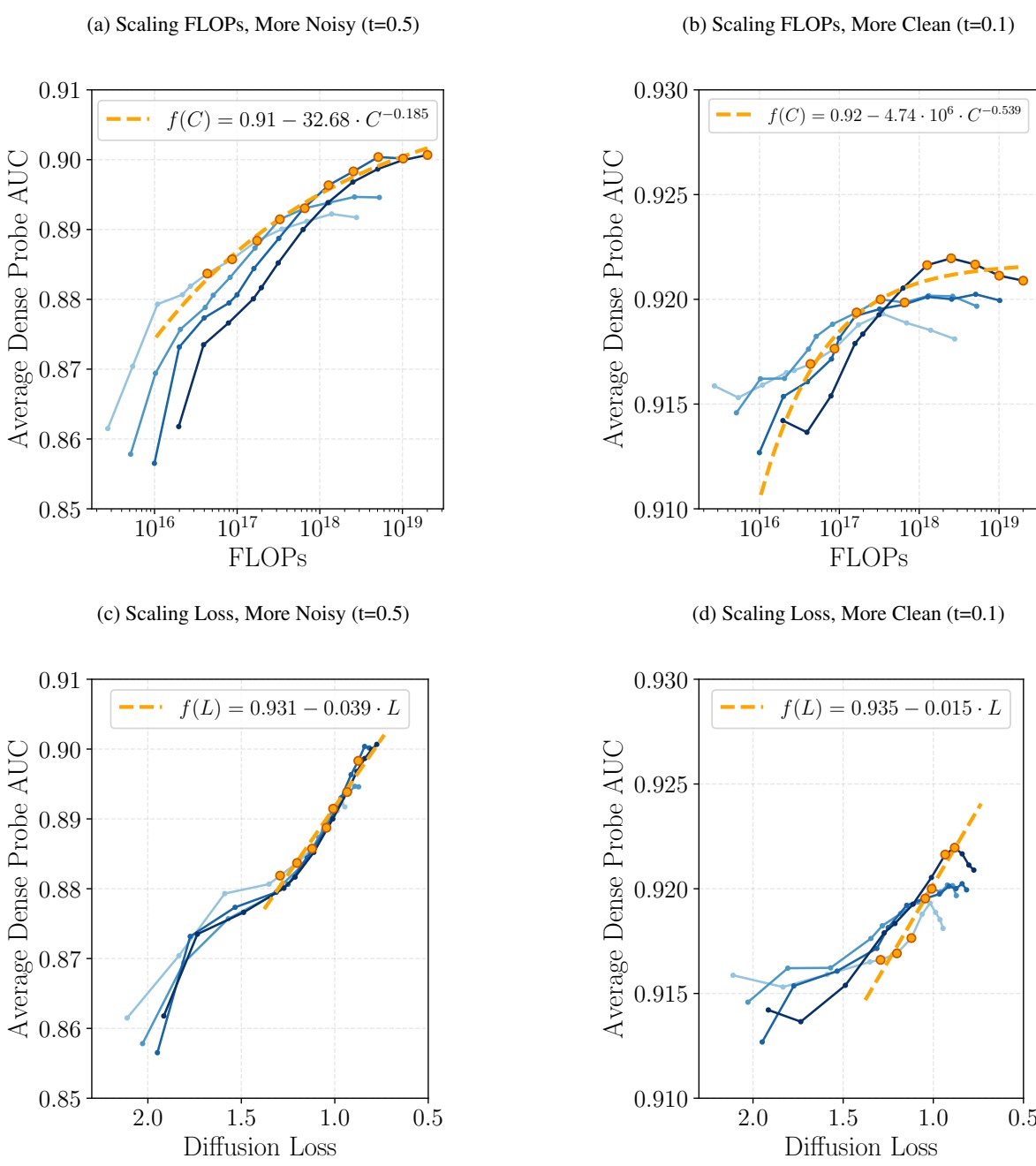

*Figure 14.* Scaling behavior of dense probing. Unlike 1-D probing, we use all the available features. Row-wise, we vary the x-axis (FLOPs vs. Diffusion Loss). Column-wise, we vary the noisiness of the diffusion input (noisy vs. clean).

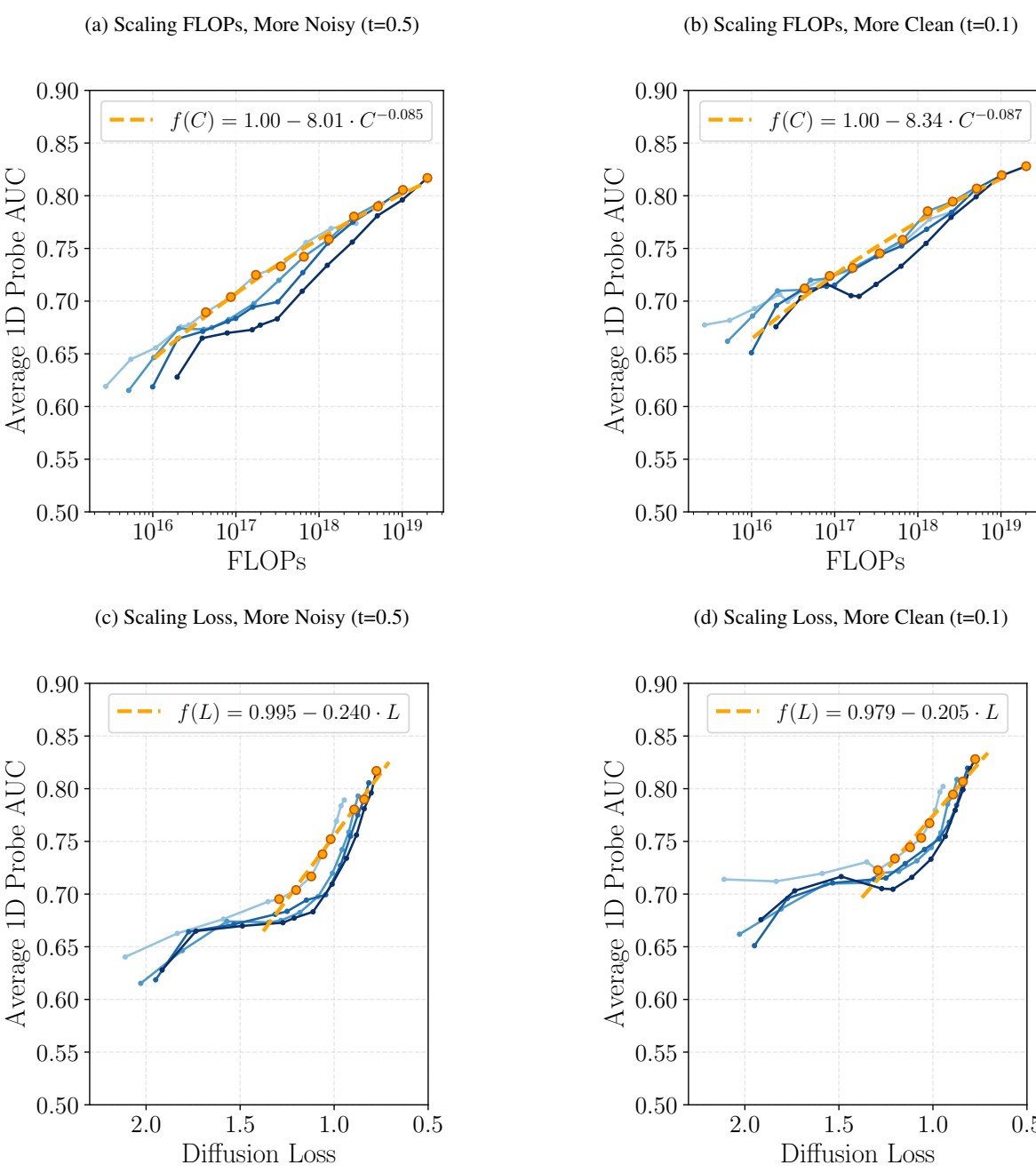

*Figure 15.* Scaling behavior of 1-D probing. Row-wise, we vary the x-axis (FLOPs vs. Diffusion Loss). Column-wise, we vary the noisiness of the diffusion input (noisy vs. clean).

Locations of Top-1 Diffusion Meta-Neurons Across Layers

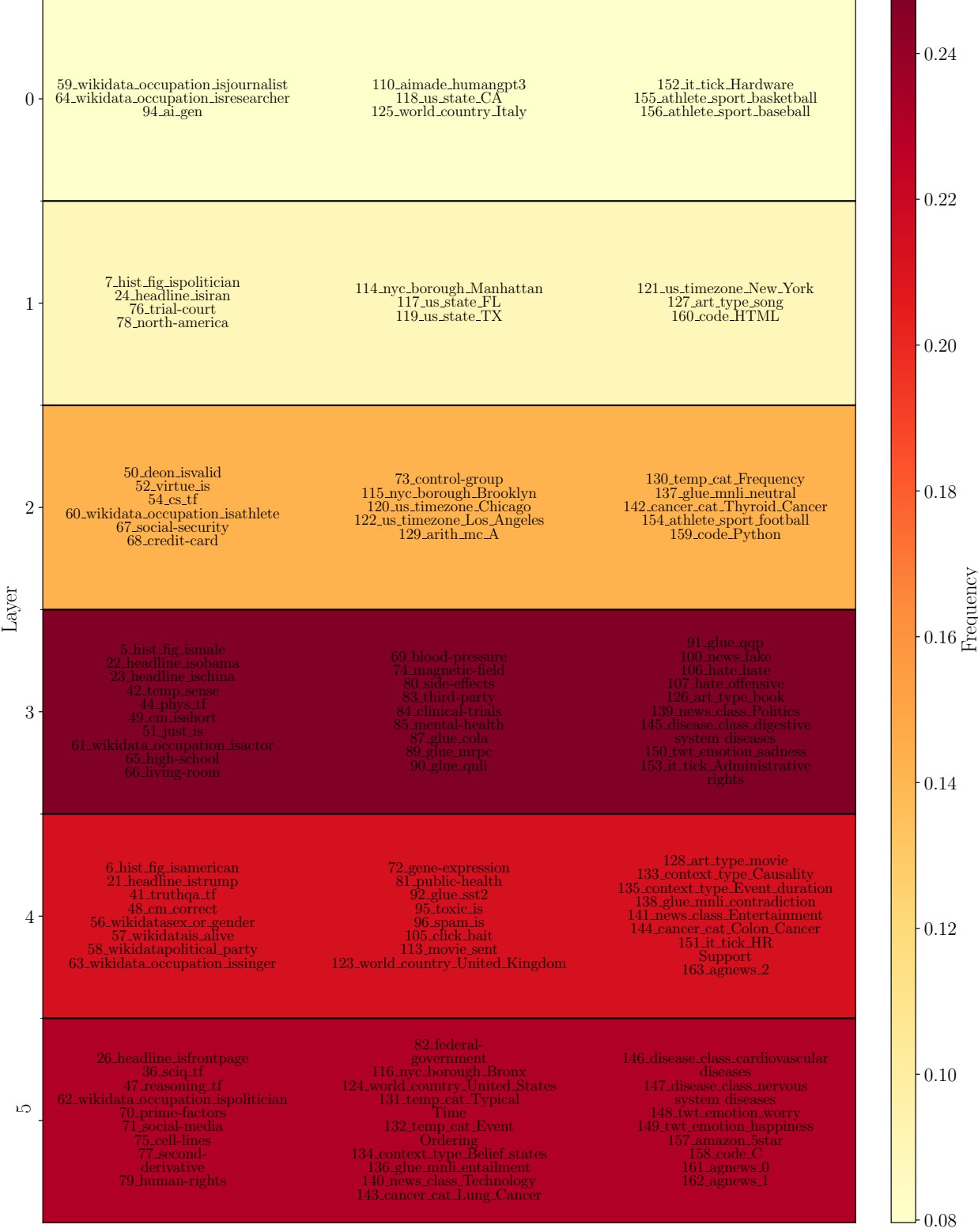

*Figure 16.* For each 1-D probing task, we depict the location of the best performing GLP meta-neuron. We also color each layer by the frequency at which it contained the best task-specific meta-neuron.

