# OpenReview forum: "Learning a Generative Meta-Model of LLM Activations"
_ICML.cc/2026/Conference — ICML 2026 regular_

### Official Review · Reviewer_tiJE · 2026-02-24

**Soundness:** 4
**Presentation:** 4
**Significance:** 3
**Originality:** 3
**Overall Recommendation:** 4
**Confidence:** 3

**Summary:**

This paper presents the Generative Latent Prior, a method for modeling neural network activations with a generative objective using diffusion models which has much weaker assumptions than approaches such as SAEs. They use an activation diffusion model on two downstream tasks: improving steering by post-processing interventions and 1D probing. They also examine the scaling performance on GLP models from 0.5B to 3.3B parameters, and find smooth loss scaling which correlates with downstream steering and probe performance.

**Compliance With Llm Reviewing Policy:**

Affirmed.

**Final Justification:**

The paper is technically solid and clearly presented, with thorough scaling analysis and a useful contribution in on-manifold steering. The rebuttal fully addressed my interpretability concern, as the monosemanticity evaluation shows GLP meta-neurons are comparable to SAE features, which strengthens the case for GLP as a practical tool.

However, my concern about the practical significance of activation steering as the headline application remains. While the authors cite growing interest, I'm not aware of meaningful public evidence of steering vectors being deployed in practice. I maintain my score of Weak Accept.

**Key Questions For Authors:**

Q1: Have you examined quantitative interpretability metrics (e.g. autointerp) for meta-neurons? I would be curious to get a better sense for the interpretability of the meta-neurons to connect with the qualitative analysis in section 5.3. If they demonstrate strong quantitative interpretability (e.g. comparable to SAE features) it would strengthen the case for GLP as a practical interpretability tool.

**Limitations:**

Yes

**Strengths And Weaknesses:**

Strengths:

The paper is well written and easy to follow. The Python-like pseudocode is a nice touch that is easy to understand.

The paper makes good use of standard interpretability evaluations from the literature (steering with LLM as a judge, delta LM loss, probing) which makes it easy to contextualize against prior work.

The use of GLP to post-process steering interventions is a nice contribution and a new tool in the toolbox.

The 1D probing results, where a GLP trained on a 1B model outperforms 8B baselines, suggest GLP is learning something meaningful.

It is nice to see exploration of activation models that are less opinionated than SAEs.

The scaling analysis is thorough, and it’s good to see a strong correlation with loss and downstream performance.

Weaknesses:

Activation steering is an active area of research, but is currently of limited practical importance and isn’t commonly deployed. This limits the impact of on-manifold steering, which is a primary contribution of the paper.

The steering improvements are most pronounced in the low fluency regime, which intuitively makes sense. In the higher fluency regime (roughly >0.6 / 60 in Figures 5 and 6), the pareto improvement is consistently there, but is often within the confidence interval, especially for the Persona Vectors.

GLP appears to be more expensive than SAEs, both for training and inference at steering time, although it’s unclear by exactly how much. However, SAEs may plateau earlier, and it can be nice to have something that can absorb more compute.

The meta-neuron analysis in Section 5.3 is nice to see. However, it would be nice to also see something like auto-interp scores in addition to the two qualitative examples to get a sense for how interpretable the meta-neurons typically are.

Overall:

I find the paper to be technically solid and clearly presented, with convincing evidence that GLP models can improve steering post-processing and produce strong single-feature probes. My main reservations are the practical impact of steering as the headline application and the (likely) higher cost relative to SAEs.

---

> ### Author Rebuttal · Authors · 2026-03-31
>
> Thank you for your support of GLP as a valuable “new tool” with “[thorough] scaling analysis.” Below we address your questions on the practicality of activation steering, and report quantitative interpretability metrics.
>
> ---
> 1. *Activation steering is an active area of research, but is currently of limited practical importance.*
>
> **Activation steering is of increasing practical interest to model providers: Persona Vectors was introduced by Anthropic [1] and described in the RLHF Book as “essential at leading AI laboratories” [2] for shaping LLM character and safety.**
>
> Our experiments show that GLP improves Persona Vectors and directly addresses a key barrier to its broader deployment: fluency degradation from off-manifold interventions. As seen in Table 1, for aggressive concept steering (threshold: 60), Persona Vectors heavily degrades (fluency: 20.6), while GLP maintains coherence (fluency: 47.2). Furthermore, Persona Vectors fundamentally cannot reach certain concept scores (denoted by “x”), while GLP extends the ceiling of concept scores from 60 to 80.
>
> **Table 1. GLP preserves fluency when steering with Persona Vectors.** For each minimum concept threshold, we report the best fluency across steering coefficients. We denote unreachable concept scores with “x”. This experiment corresponds to Figure 6a of the main paper.
> | Concept Threshold | Persona Vector  - Fluency (↑) | +GLP  - Fluency (↑) |
> |-|-|-|
> | 10 | 68.7 | **70.2** |
> | 20 | 58.1 |  **63.4** |
> | 30 | 47.3 |  **56.8** |
> | 40 | 41.5 |  **47.2** |
> | 50 | 41.5 |  **47.2** |
> | 60 | 20.6 |  **47.2** |
> | 70 | x | **34.0** |
> | 80 | x | **19.6** |
>
> [1] Chen et. al. Persona Vectors: Monitoring and Controlling Character Traits in Language Models. Anthropic Blog, 2025. https://www.anthropic.com/research/persona-vectors. \
> [2] Lambert. RLHF Book: Product, UX, and Model Character, 2026. https://rlhfbook.com/c/17-product.
>
> ---
> 2. *In the higher fluency regime [...] the pareto improvement is consistently there, but is often within the confidence interval.*
>
> We view this as expected behavior rather than a limitation, as GLP helps when needed (in the low fluency regime) and does not interfere when less needed (in the high fluency regime). In the high fluency regime, activations are already near-manifold, so there is less need for correction.
>
> ---
> 3. *Have you examined quantitative interpretability metrics [...] for meta-neurons?*
>
> We conduct a quantitative interpretability evaluation following the LLM-as-a-judge protocol from He et al. [3]:
>
> **Table 2. Monosemanticity scores (1-5 scale; higher = more monosemantic).** An LLM judge grades whether the feature activates on a single coherent concept, based on top-20 max-activating examples across 1000 randomly sampled features per method. Methods correspond to the Llama8B variants in Table 4 of the main paper. **GLP meta-neurons are indeed comparable to SAE features, indicating its promise as a practical interpretability tool.**
>
> | Method | Monosemanticity (↑) | 95% CI |
> |--------|-----------------|--------|
> | Raw Layer Output | 3.948 | [3.909, 3.987] |
> | Raw MLP Neuron | 4.208 | [4.170, 4.245] |
> | SAE | 4.241 | [4.202, 4.279] |
> | GLP | 4.265 | [4.226, 4.302] |
>
> [3] He et. al. Llama Scope: Extracting Millions of Features from Llama-3.1-8B with Sparse Autoencoders.

---

> > ### Author Rebuttal · Reviewer_tiJE · 2026-04-03
> >
> > Thank you for the monosemanticity evaluation, which addresses my interpretability question.
> >
> > On practical importance: I read the RLHF Book section cited in the rebuttal. Among the methods discussed there, character training and model specifications both have significant public evidence of being used in practice (e.g., company blog posts, podcast discussions, https://www.anthropic.com/research/claude-character, https://model-spec.openai.com/2025-12-18.html). However, I'm not aware of comparable public evidence for steering vectors. Thus I maintain my score.

---

### Official Review · Reviewer_2kHn · 2026-03-04

**Soundness:** 3
**Presentation:** 3
**Significance:** 2
**Originality:** 3
**Overall Recommendation:** 4
**Confidence:** 3

**Summary:**

+ Formulates a Generative Latent Prior (GLP) using continuous flow-matching diffusion directly on LLM residual stream activations. Bypasses static linear and sparse inductive biases of SAEs by modeling the unconstrained intrinsic distribution of internal network states. Processes 1 billion tokens using deep Transformer-based SwiGLU MLPs to approximate target velocity for data-to-noise interpolation.
+ Establishes predictive power-law scaling parameters mapping computational investment to diffusion loss reduction and downstream interpretability utility. Demonstrates quantitative superiority in Representation Frechet Distance (Rep FD) compared to highly generous SAE baselines. Validates zero-shot transferability of continuous generative manifolds between base and instruction-tuned model variants.
+ Validates the unconstrained prior formulation via practical interpretability implementations, specifically on-manifold steering and single-scalar feature probing. Integrates SDEdit-style sampling to dynamically correct off-manifold artifacts introduced by linear concept vectors. Deploys internal diffusion meta-neurons as unsupervised feature encoders that outperform L1-regularized dictionary learning baselines.

**Compliance With Llm Reviewing Policy:**

Affirmed.

**Final Justification:**

I maintain my positive score.

**Key Questions For Authors:**

1. What is the exact wall-clock latency multiplier when applying the 20-step GLP correction algorithm during active autoregressive generation compared to a standard linear SAE addition?
2. How sensitive is the Llama8B perplexity degradation metric to fluctuations in the noise threshold hyperparameter ($t_{start}$), and does the claimed outperformance over SAEs hold at higher levels of geometric corruption?
3. Given the severe efficiency degradation noted in Appendix B.1 for simply scaling across layers, what theoretical modifications to the flow-matching objective are required to model multi-token attention matrices without compute explosion?
4. Have formal ablation studies been conducted to guarantee top-1 meta-neurons are truly monosemantic, or do these units simply exploit L2-regularized logistic regression to mask concurrent activation on unrelated high-frequency concepts?
5. Minor issue: it could be better using flow matching rather than diffusion for that you are using flow matching.

**Limitations:**

Status: Adequately Discussed.

Suggestion: Provide explicit documentation of the exact wall-clock inference latency and VRAM spikes required to execute multi-step SDE dit-style sampling during autoregressive generation to clarify practical deployment feasibility.

**Strengths And Weaknesses:**

Strengths:
1. Paradigm shift transitioning from rigid linear structural priors to continuous generative meta-modeling of high-dimensional transformer states. Elegantly adapts computer vision continuous flow-matching techniques into natural language internal representation spaces. Circumvents permutation symmetry problems of weight-space modeling by strictly generating dynamic residual stream activations.
2. Rigorous statistical validation of unconditional generation quality using Rep FD benchmarks mirroring visual FID paradigms. Solidifies evaluations by calculating irreducible finite-sample lower bounds to properly contextualize GLP convergence limits. Successfully contextualizes performance against highly generous SAE reconstructions initialized directly from actual training activations.
3. Derives clear, predictive power-law scaling mapping FLOP computational investment directly to enhanced interpretability. Quantifies the exact FLOP-to-utility exchange rate, demonstrating that minimizing the unsupervised diffusion objective intrinsically refines complex downstream semantic boundaries. Provides a highly robust, scalable operational blueprint for future foundation model alignment.

Weakness:
1. Delta LM Loss evaluation deploys confounding variables, rendering direct juxtaposition against Sparse Autoencoders mathematically asymmetrical and highly questionable. Repurposes a metric designed for static dictionary bottlenecks by injecting arbitrary temporal noise variables based strictly on a manually selected threshold ($t_{start} = 0.5$). Obscures true information retention capabilities by comparing multi-step noise estimation against strict linear projections.
2. Exclusion of multi-token sequence dynamics completely ignores the fundamental nature of autoregressive causal language modeling. Discards complex attention mechanisms, contextual feature circuits, and sequential reasoning geometries by strictly limiting the architecture to isolated single-token representations. Risks heavily biasing the meta-model toward shallow syntactic features rather than deep semantic reasoning chains.
3. Critically omits real-world inference latency and computational overhead required for active autoregressive deployment. Fails to quantify the massive computational disparity between adding a static linear vector versus executing a 20-step differential equation sampling process per generated token. Masks the severe operational infeasibility of running dynamic statistical initializations during live generation.
4. Relies exclusively on 113 simplistic binary classification tasks for 1-D scalar probing, risking severe overstatement of GLP feature disentanglement. Fails to evaluate highly abstract, polysemantic, or continuous variables against the dense 131k-feature SAE dictionaries. Masks potential polysemanticity by allowing extensive L2 regularization tuning on dense output layers to easily extract binary concepts from entangled meta-neurons.

---

> ### Author Rebuttal · Authors · 2026-03-31
>
> Thank you for your positive assessment of our work as “a highly robust, scalable operational blueprint for future foundation model alignment” with “rigorous statistical validation.” Below, we address your questions on runtime, the perplexity degradation metric, multi-token attention modeling, and monosemanticity.
>
> ---
> 1. *What is the exact wall-clock latency multiplier when applying the 20-step GLP correction algorithm [...] compared to a standard linear SAE addition?*
>
> We report the wall-clock latency multiplier of GLP vs. SAE in Table A below.
>
> **Table A. Wall-clock latency of linear SAE addition vs. 20-step GLP correction.** We evaluate on Llama1B, for a single token, averaging over 100 runs.
> | SAE (↓) | +GLP (↓)  | Wall-Clock Multiplier |
> |-|-|-|
> | 0.0154 s | 0.0312 s | 2.03x |
>
> We note that our algorithm applies a nonlinear transformation to denoise the steered direction, whereas the SAE baseline applies a linear transformation. This additional expressivity achieves large gains in steering performance, as shown in Figure 5 in the main paper. Given our algorithm’s greater performance and the absolute latency multiplier being small, our algorithm is both beneficial and feasible to run during live generation.
>
> ---
> 2. *How sensitive is the Llama8B perplexity degradation metric to fluctuations in the noise threshold hyperparameter, and does the claimed outperformance over SAEs hold at higher levels of geometric corruption?*
>
> In Table B below, we ablate the noise hyperparameter and validate that GLP outperforms SAEs in the perplexity degradation metric (Delta LM Loss), even at high corruption levels.
>
> **Table B. Delta LM Loss for SAE vs. GLP, for inputs of varying noise levels.** This experiment corresponds to the Llama8B-Base setting in Table 2 of the main paper.
>
> | Noise Level | SAE (↓) | +GLP (↓)  |
> |-|-|-|
> | t=0.0 (no noise) | 0.1976 |  **0.0000** |
> | t=0.5 (medium noise) | 4.8862 | **0.0513** |
> | t=0.6 | 6.2626 | **0.1093** |
> | t=0.7 | 6.8582 | **0.2151** |
> | t=0.8 | 6.9589 | **0.5684** |
> | t=0.9 (high noise) | 7.0314 | **4.9345** |
> ---
> 3. *Given the severe efficiency degradation noted in Appendix B.1 for simply scaling across layers, what theoretical modifications to the flow-matching objective are required to model multi-token attention matrices without compute explosion?*
>
> No modifications to the flow-matching objective would be needed. **Existing image generation methods like DiT [1] and SiT [2] demonstrate that diffusion and flow-matching scale efficiently over multi-token sequences under standard compute budgets.** To clarify, the multi-layer and multi-token setups involve different architectures (AdaLN [1] vs. Attention [3]); one cannot infer the scaling behavior from one to the other.
>
> We opted for single-token modeling to maintain comparability with SAEs. Multi-token sequence dynamics is an exciting area of future work currently overlooked by the field as a whole. Our primary goal in this work is to validate the viability of flow-matching compared with prior activation modeling methods, which we demonstrate via superior steering and probing results.
>
> [1] Peebles et. al. Scalable Diffusion Models with Transformers. ICCV 2023. \
> [2] Ma et. al. SiT: Exploring Flow and Diffusion-based Generative Models with Scalable Interpolant Transformers. ECCV 2024. \
> [3] Vaswani et. al. Attention Is All You Need. NeurIPS 2017.
>
> ---
> 4. *Have formal ablation studies been conducted to guarantee top-1 meta-neurons are truly monosemantic?*
>
> We conduct a formal ablation study for monosemanticity following the LLM-as-a-judge protocol from He et al. [4]:
>
> **Table C. Monosemanticity scores (1-5 scale; higher = more monosemantic).** An LLM judge grades whether the feature activates on a single coherent concept, based on top-20 max-activating examples across 1000 randomly sampled features per method. Methods correspond to the Llama8B variants in Table 4 of the main paper. **GLP meta-neurons are indeed as monosemantic as SAE features, as shown through the above quantitative evaluation.**
>
> | Method | Monosemanticity (↑) | 95% CI |
> |--------|-----------------|--------|
> | Raw Layer Output | 3.948 | [3.909, 3.987] |
> | Raw MLP Neuron | 4.208 | [4.170, 4.245] |
> | SAE | 4.241 | [4.202, 4.279] |
> | GLP | 4.265 | [4.226, 4.302] |
>
> [4] He et. al. Llama Scope: Extracting Millions of Features from Llama-3.1-8B with Sparse Autoencoders.
>
> ---
> 5. *Minor issue: it could be better [to use the term] flow matching rather than diffusion.*
>
> We indeed use flow matching and will clarify this point in the final manuscript.

---

> > ### Author Rebuttal · Reviewer_2kHn · 2026-04-01
> >
> > Thank you for your reply, and i maintain my positive score.

---

### Official Review · Reviewer_7bKm · 2026-03-07

**Soundness:** 3
**Presentation:** 3
**Significance:** 3
**Originality:** 3
**Overall Recommendation:** 4
**Confidence:** 2

**Summary:**

This paper introduces the Generative Latent Prior (GLP), a novel approach that uses diffusion models to learn the distribution of LLM residual stream activations. Unlike existing methods like PCA or Sparse Autoencoders (SAEs) that rely on strong structural assumptions (e.g., linearity or sparsity), GLP makes minimal assumptions and scales effectively with data and model size. The study evaluates GLP models ranging from 0.5B to 3.3B parameters trained on one billion activations, showing that diffusion loss serves as a reliable proxy for downstream performance.

**Compliance With Llm Reviewing Policy:**

Affirmed.

**Final Justification:**

Concerns are solved.

**Key Questions For Authors:**

see weakness

**Limitations:**

see weakness

**Strengths And Weaknesses:**

Strength: The paper shifts the paradigm from restrictive structural models (like SAEs) to a flexible generative framework using diffusion, which better captures the complex, non-linear manifold of LLM activations.


Weakness: While the paper demonstrates the theoretical advantages of using a diffusion-based generative prior over linear models like SAEs, the practical real-world utility of deploying a multi-billion parameter meta-model solely for activation monitoring remains unclear. (I am not familar with this topic). Could the authors provide a more concrete analysis or examples?

---

> ### Author Rebuttal · Authors · 2026-03-31
>
> Thank you for your positive comments recognizing GLP as a “paradigm [shift]” towards a “flexible generative framework,” and for identifying key points to clarify. Below we provide more concrete analysis and examples of GLP’s practical utility.
>
> ---
>
> 1. *[T]he practical real-world utility [...] remains unclear.*
>
> **GLP pushes the frontier of activation steering, which is of increasing practical interest to model providers. Persona Vectors was introduced by Anthropic [1] and described in the RLHF Book as “essential at leading AI laboratories” [2] for shaping LLM character and safety.**
>
> Our experiments show that GLP improves Persona Vectors and directly addresses a key barrier to its broader deployment: fluency degradation from off-manifold interventions. As seen in Table 1, for aggressive concept steering (threshold: 60), Persona Vectors heavily degrades (fluency: 20.6), while GLP maintains coherence (fluency: 47.2). Furthermore, Persona Vectors fundamentally cannot reach certain concept scores (denoted by “x”), while GLP extends the ceiling of concept scores from 60 to 80.
>
> **Table 1. GLP preserves fluency when steering with Persona Vectors.** For each minimum concept threshold, we report the best fluency across steering coefficients. This experiment corresponds to Figure 6a of the main paper.
> | Concept Threshold | Persona Vector  - Fluency (↑) | +GLP  - Fluency (↑) |
> |-|-|-|
> | 10 | 68.7 | **70.2** |
> | 20 | 58.1 |  **63.4** |
> | 30 | 47.3 |  **56.8** |
> | 40 | 41.5 |  **47.2** |
> | 50 | 41.5 |  **47.2** |
> | 60 | 20.6 |  **47.2** |
> | 70 | x | **34.0** |
> | 80 | x | **19.6** |
>
> [1] Chen et. al. Persona Vectors: Monitoring and Controlling Character Traits in Language Models. Anthropic Blog, 2025. https://www.anthropic.com/research/persona-vectors. \
> [2] Lambert. RLHF Book: Product, UX, and Model Character, 2026. https://rlhfbook.com/c/17-product.

---

> > ### Author Rebuttal · Reviewer_7bKm · 2026-04-02
> >
> > Thank you for your explaination

---

### Official Review · Reviewer_LvwR · 2026-03-10

**Soundness:** 4
**Presentation:** 3
**Significance:** 3
**Originality:** 3
**Overall Recommendation:** 5
**Confidence:** 2

**Summary:**

The authors propose training a generative model (diffusion or flow matching) to learn the distribution of activations of a given layer of an LLM. Once trained, this models learns the manifold of activations within this layer and can be used for some interesting applications. For example, pushing an LLM into a desired direction risks pushing the activations off the manifold they are typically on. But if the manifold is learned by the generative model, that model can be used as a corrective prior. I.e., first push the model in the desired direction, then use the diffusion model to move the activations to the correct manifold in the same style as done in vision methos like SDEdit. The same diffusion model can also b used to sharpen SAE or be used for simple 1D probing for interpretability. In practice, the authors use the middle layer of an LLM and perform experiments with Llama1B and Llama8b.

**Compliance With Llm Reviewing Policy:**

Affirmed.

**Final Justification:**

The authors rebuttal answers my questions and I maintain my positive review. Although not a breakthrough (hence ACCEPT instead of STRONG ACCEPT), I do think this is a strong paper and the community will benefit from its publication. I am happy that this is mostly a unanimous view among all reviewers.

**Key Questions For Authors:**

line 153, left column: " For simplicity we choose to model single-token rather than multi-token activations," and also line 149, right column: " we start by training unconditional GLPs". Future directions hints at fine-tuning conditional models. Conditioned on what? on the query? or? It would actually be a very interesting analysis to see if and by how much the manifold changes based on query topics.

**Limitations:**

Overall, this is an interesting paper. The idea is simple and as the authors note done in the vision community. Nevertheless, applying it to LLMs and also characterizing the scaling laws are quite good. I don't see any major limitations except I have a quetions.

line 107, right column: " For simplicity we choose to model single-token rather than multi-token activations". I would like to know the reason for this choice and its effect on the model. Perhaps a future direction could be multi-token activations. But for now, a short discussion or explanation as to what this design choice means would be good.

**Strengths And Weaknesses:**

This is a simple yet effective idea. As the authors note, this has been done in computer vision but, perhaps surprisingly, not yet on LLMs. Results on LLM steering (to stay on manifold), improving SAE, and eliciting personas are good. Another strong point of the paper is that the authors characterize the scaling laws and demonstrate smooth power-law scaling.

---

> ### Author Rebuttal · Authors · 2026-03-31
>
> Thank you for the thoughtful discussion, and for highlighting GLP as a “simple yet effective idea” with “smooth power-law scaling.” We address your questions below.
>
> ---
>
> 1. *I would like to know the reason for [single-token rather than multi-token modeling] and its effect.*
>
> We opted for single-token modeling primarily for comparability with SAEs, which are traditionally formulated as shallow MLPs without attention. This means that both methods handle each token position independently, rather than considering the joint distribution across positions. We agree that multi-token activations are an exciting future direction, as they could reveal new findings like meta-neurons that capture phrase-level concepts.
>
> ---
>
> 2. *[What would future] conditional models [...] [be conditioned on]?*
>
> Indeed, conditioning on the topic of the query prompt would be quite interesting.
> The unconditional model already exhibits some topic-specific structure in Sec. 5 of the main paper (e.g., a meta-neuron that distinguishes baseball vs. non-baseball related queries), and a conditional model explicitly conditioned on ground-truth topic labels could learn this structure more precisely or discover more fine-grained topics.

---

> > ### Author Rebuttal · Reviewer_LvwR · 2026-04-01
> >
> > Thank you for your clarifications. I maintain my positive score and I am happy that other reviewers also assessed the paper positively.

---

### Decision · Program_Chairs · 2026-04-30

**Decision:**

Accept (regular)

**Comment:**

All reviewers voted for acceptance, confirming that their concerns were fully resolved. Reviewers praised the paper for its technical rigor, clarity of presentation, thorough scaling analysis, and practical contributions. The consensus is that this is a strong paper from which the community will benefit significantly.